# New botulinum neurotoxin constructs for treatment of chronic pain

Charlotte Leese[1],*, Claire Christmas[1],*, Judit Mészáros[1], Stephanie Ward[1], Maria Maiaru[3], Stephen P Hunt[2], Bazbek Davletov[1,4]

**Chronic pain affects one in five people across human societies, with few therapeutic options available. Botulinum neurotoxin (BoNT) can provide long-lasting pain relief by inhibiting local release of neuropeptides and neurotransmitters, but its highly paralytic nature has limited its analgesic potential. Recent advances in protein engineering have raised the possibility of synthesising non-paralysing botulinum molecules for translation to pain sufferers. However, the synthesis of these molecules, via several synthetic steps, has been challenging. Here, we describe a simple platform for safe production of botulinum molecules for treating nerve injury–induced pain. We produced two versions of isopeptide-bonded BoNT from separate botulinum parts using an isopeptide bonding system. Although both molecules cleaved their natural substrate, SNAP25, in sensory neurons, the structurally elongated iBoNT did not cause motor deficit in rats. We show that the non-paralytic elongated iBoNT targets specific cutaneous nerve fibres and provides sustained pain relief in a rat nerve injury model. Our results demonstrate that novel botulinum molecules can be produced in a simple and safe manner and be useful for treating neuropathic pain.**

## Introduction

Botulinum neurotoxins (BoNTs) are important biopharmaceuticals for the treatment of muscular and secretory conditions. These toxins potently silence neuromuscular junctions (NMJs) for several months but can also block neurotransmitter release from other types of neurons (Pirazzini et al, 2017). Botulinum neurotoxin type A (BoNT/A) is now widely used in clinical and aesthetic medicine because of its long-lasting nerve block. Injecting BoNT/A at picogram to low nanogram amounts causes muscle relaxation lasting up to 5 mo (Pirazzini et al, 2017). BoNT/A injections have also attracted attention as a possible long-lasting and non-addictive method of pain relief (Attal et al, 2016; Zhang et al, 2016; Luvisetto,

2021); however, its paralytic properties present a substantial obstacle to such applications. Novel long-lasting analgesics are urgently required because chronic and persistent pain is a major health problem estimated to affect over 20% of the human population (Borsook et al, 2014; Fayaz et al, 2016), leading to many socioeconomic problems including the opioid crisis (Phillips, 2009). Biopharmaceutical production of BoNT/A, the most toxic substance known to humans, requires great care and is strictly regulated. Therefore, only a dozen companies worldwide are authorised to produce BoNT/A as a biopharmaceutical medicine. The highly toxic nature of BoNTs impedes progress in the field of novel botulinum therapeutics and necessitates new approaches to innovation.

Structurally, BoNT/A is a large 150-kD protein consisting of a 50-kD enzymatic SNAP25-cleaving light chain (L) linked by a disulphide bond to a 100-kD heavy chain, consisting of translocation and binding domains (Fig 1A) (Montal, 2010). The binding domain, namely, the C-terminal half of the heavy chain (HC), binds to neuronal receptors—ganglioside GT1b and synaptic vesicle protein 2 (SV2), the latter being exposed after release of neurotransmitters and before recovery by endocytosis (Montal, 2010; Surana et al, 2018). After binding of its neuronal receptors, BoNT/A hijacks endocytosis to enter nerve terminals. Once in the endocytic vesicle, the translocation domain (the N-terminal half of the heavy chain, HN) forms a pore in the vesicular membrane allowing the light chain to enter the cytosol (Montal, 2010). The free light chain then cleaves the SNAP25 protein involved in vesicle fusion resulting in a long-lasting nerve block (Schiavo et al, 1993; Binz et al, 1994).

We describe here a new system for the safe manufacturing of botulinum molecules from two separately produced parts. SpyCatcher technology is a bonding technique, which allows covalent linking of proteins (Zakeri et al, 2012) via an isopeptide bond between a small protein called SpyCatcher and a 12–amino acid peptide called SpyTag. This covalent peptide bonding can be used for bioconjugation, as fusion of the SpyCatcher and SpyTag to two independent proteins allows linking of these proteins by simple mixing. We show here that the SpyCatcher–SpyTag technology can be used to effectively bond two parts of BoNT to generate novel

---

[1]Department of Biomedical Science, University of Sheffield, Sheffield, UK [2]Cell and Developmental Biology, University College London, London, UK [3]Department of Pharmacology, School of Pharmacy, University of Reading, Whiteknights Campus, Reading, UK [4]Neuresta, Inc., San Diego, CA, USA

Correspondence: b.davletov@sheffield.ac.uk
*Charlotte Leese and Claire Christmas contributed equally to this work

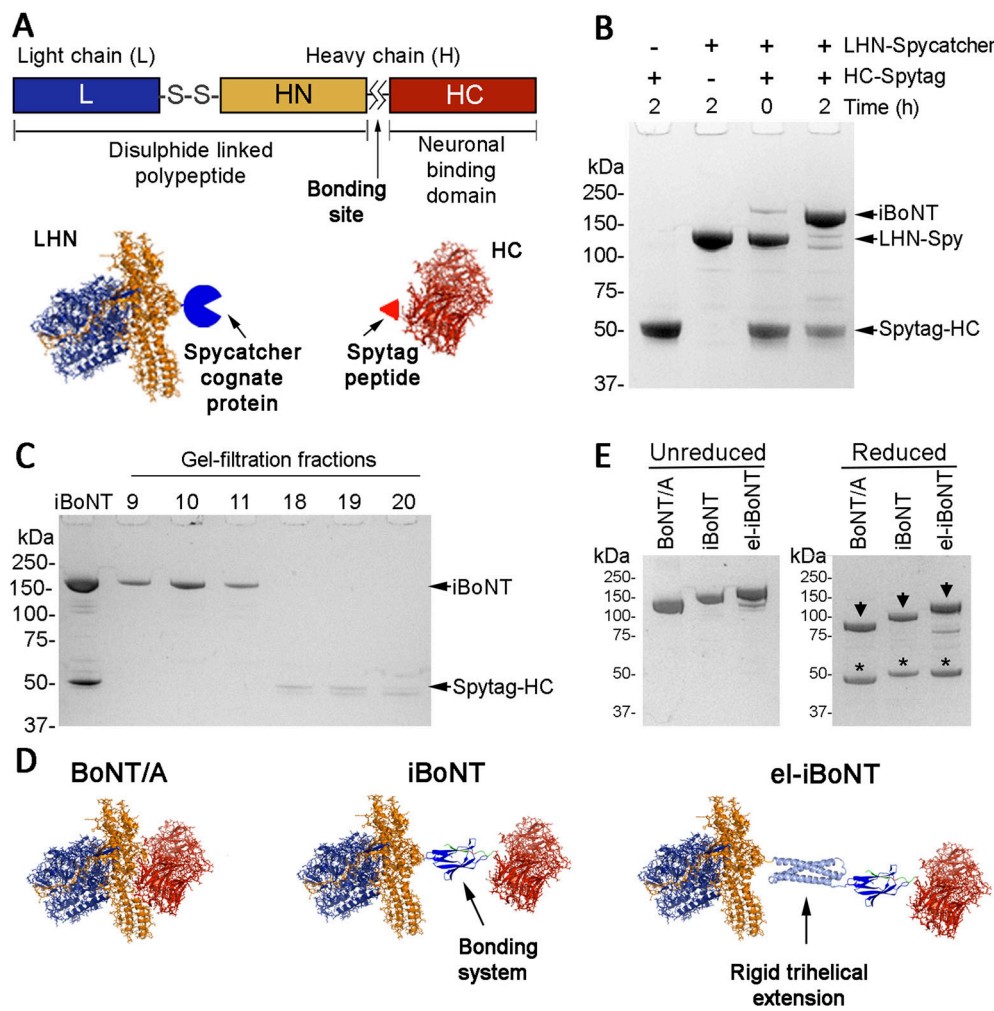

**Figure 1. Production of isopeptide-bonded BoNT molecules.**
**(A)** Schematic of the structural organisation of BoNT/A (upper panel) with additions of the SpyCatcher elements to two botulinum parts (lower panel). **(B)** Coomassie-stained SDS–PAGE showing the purified proteins and spontaneous assembly of isopeptide-bonded BoNT. **(C)** Coomassie-stained SDS–PAGE showing isolation of iBoNT from the unreacted excess of SpyTag–HC by gel filtration. **(D)** Structural models of native BoNT/A (left), non-elongated iBoNT (centre), and elongated iBoNT (right) molecules based on known crystal structures (Lacy et al, 1998; Lerman et al, 2000; Li et al, 2014). **(E)** Coomassie-stained SDS–PAGE demonstrating increasing molecular weights of iBoNT molecules compared with the native BoNT/A molecule, in non-reducing (left) and reducing conditions (right). The light chains (asterisks) become separated from the heavy chains (arrowheads) upon reduction in the critical disulphide bond.

nerve blockers, referred to as isopeptide-bonded BoNT (iBoNT) herein. We demonstrate that a structurally elongated iBoNT (el-iBoNT) construct has a reduced ability to penetrate NMJs and therefore exhibits markedly reduced paralytic properties. Because long-lasting nerve blockade in the absence of muscle paralysis could be beneficial for treating chronic pain conditions, we investigated el-iBoNT in a well-recognised model of nerve injury pain. Our data show that el-iBoNT can treat chronic neuropathic pain without adverse effects, making it a promising candidate for human clinical research.

## Results

### Isopeptide-bonded botulinum molecules

For simplicity, the BoNT structure can be described as LHN–HC where LHN is a tightly organised light-chain/translocation unit, responsible for entry into the cytosol and SNAP25 cleavage. We fused LHN (aa 1–873) to SpyCatcher (LHN–SpyCatcher), whereas the neuronal binding domain of BoNT/A (HC, aa 874–1296) was fused to SpyTag–HC (Fig 1A). Both proteins were expressed in *E. coli* and purified to homogeneity (Fig 1B). Combining the two parts led to

formation—within 2 h—of a new protein called iBoNT (Fig 1B). Further gel filtration in 100 mM NaCl and 20 mM Hepes, pH 7.3 (buffer A), resulted in separation of the iBoNT from the small excess of SpyTag–HC (Fig 1C). Stability of iBoNT was evident from its resistance to harsh SDS–PAGE conditions (Fig 1B and C) and its ability to stay intact in solution up to 72 h at 4°C (data not shown). Using isopeptide bonding, we then prepared an el-iBoNT (Fig 1D and E) where we inserted a trihelical domain from syntaxin 1A (aa 2–160) between the LHN and SpyCatcher (Fig 1D). A similarly elongated, but not covalently bonded, botulinum molecule, Bitox, was shown previously to have analgesic properties in the absence of muscle paralysis (Fig S1A) (Ferrari et al, 2011; Mangione et al, 2016). After protein expression and purification of the covalently bonded el-iBoNT (Fig S2), we analysed the relative molecular weights and stability of iBoNT molecules by SDS–PAGE. As expected, el-iBoNT was stable and, with a MW of 182,255, migrated slower than the non-el-iBoNT (MW 168,773) or native BoNT/A (MW 149,425) (Fig 1D and E).

### Testing functional activity of isopeptide-bonded BoNTs

We investigated the SNAP25-cleaving ability of engineered botulinum molecules in neuronal cultures with native BoNT/A as a

positive control. Cleavage of SNAP25 was evidenced by a shift in its electrophoretic mobility observed in immunoblots. Fig 2A shows that BoNT/A and both iBoNT molecules cleaved SNAP25 in the low picomolar range after 68-h application to rat dorsal root ganglion (DRG) neurons. No cleavage of SNAP25 could be observed when LHN–SpyCatcher molecules were applied alone to neuronal cultures in the picomolar range (Fig S3), confirming that the binding domain is required for botulinum activity in neurons. In addition, immunostaining of DRG cultures treated with native BoNT/A or iBoNT for 24 h showed a similar degree of SNAP25 cleavage in medium- to large-size neurons (Fig 2B, right) as revealed by a rabbit antibody developed specifically to recognise the SNAP25 cleavage epitope TRIDEANQ (Fig 2B, left) (Andreou et al, 2021).

We then investigated the paralytic activity of isopeptide-bonded BoNT molecules using electromyography (EMG) to allow an assessment of muscle function in rats after subcutaneous injections. Lightly anaesthetised animals were injected subcutaneously with non-elongated or el-iBoNT above the gastrocnemius muscle, and muscle activity in response to electrical stimulation was recorded on day 3, when BoNTs exert their full SNAP25-cleaving activity and hence the paralytic action. Quantitative EMG analysis revealed that animals injected with the non-el-iBoNT exhibited significant motor deficit, whereas in the case of el-iBoNT, the motor function did not differ significantly from the starting value (Fig 2C). Immunohistochemical analysis of botulinum activity on ipsilateral gastrocnemius muscle was conducted using the antibody against the cleaved end of SNAP25. Fig 2D shows stronger cleavage of SNAP25 in bungarotoxin-labelled NMJs in the case of non-el-iBoNT compared with el-iBoNT, indicating that the elongated molecule does not function well in NMJs. When iBoNT molecule (20 ng) was injected directly into the gastrocnemius muscle of adult rats, a characteristic splaying of the injected leg (indicating local paralysis) was observed in the non-el-iBoNT group, whereas rats injected with el-iBoNT moved normally without any signs of motor deficit (Fig S4A, B, and E). As expected, injections of individual, unlinked botulinum parts, even at 100 ng, into the gastrocnemius muscle did not impair the motor behaviour of rats (Fig S4C and D).

## Analgesic action of el-iBoNT

A non-paralysing long-lasting nerve blocker would be ideal for alleviation of chronic pain syndromes. The therapeutic effect of non-paralytic el-iBoNT was evaluated in a rat spared nerve injury (SNI) model of neuropathic pain (Decosterd & Woolf, 2000). Rats were anaesthetised and the common peroneal and tibial nerves ligated and sectioned in one leg. All animals undergoing the SNI procedure exhibited sustained mechanical hypersensitivity (Fig 3A), evidenced by guarding of the injured leg. We have previously shown that the peak mechanical hypersensitivity is already reached by 3 d after SNI (Mangione et al, 2016). To test the analgesic action of the new constructs, rats were administered with a single injection of el-iBoNT (1, 5, 10, and 20 ng in 30 µl) or the vehicle (0.5% human serum albumin) into the lateral plantar surface of the ipsilateral hind paw on day 4 after SNI. Behavioural and paw withdrawal changes were then analysed over 23 d using the up-and-down von Frey method (Chaplan et al, 1994). Observational analysis revealed that animals injected with vehicle showed reluctance to place

ipsilateral hind paw onto the mesh surface of the behavioural apparatus and showed reduced weight bearing on the affected limb (Fig S5). In contrast, animals injected with el-iBoNT started making contact more readily with the mesh flooring after 3 d and exhibited more normal gait behaviours after a week. The el-iBoNT–injected animals demonstrated a significant reduction in mechanical hypersensitivity compared with the vehicle ($P < 0.01$) as observed at day 10 post-injection for the 5-, 10-, and 20-ng groups. Importantly, at day 24 post-injection, the el-iBoNT–treated animals even at 5 ng showed mechanical withdrawal thresholds comparable to the sham-operated group, suggesting an extensive and sustained recovery. The immunohistochemical studies performed within dorsal root ganglia and spinal cord tissues showed a lack of cleaved SNAP25, indicating no evidence of its translocation to DRG or spinal cord or a systemic effect after local injections (Fig S6A and B). Furthermore, there was no effect on mechanical (von Frey) thresholds after injections of el-iBoNT in uninjured rats, which exhibited normal behaviours throughout the experiment (data not shown). We conclude that el-iBoNT alleviates nerve injury–induced mechanical hypersensitivity in injured rats, without observable side effects.

To gain further insights into the pain-alleviating effects of el-iBoNT, we investigated the state of microglial activation in the dorsal horn of the spinal cord after induction of SNI. It is well established that spinal microglia become activated after nerve injury and they participate in the pain chronification process (Romero-Sandoval et al, 2008; Zhao et al, 2017). Indeed, after nerve injury, we found a significant increase in recruitment of spinal microglia in the ipsilateral dorsal horn as evidenced by increased Iba-1 immunostaining compared with sham-operated animals (Fig 3B). Intraplantar injection of el-iBoNT at 20 ng, however, resulted in a reduction in microglial activation (Fig 3B). This result indicates that peripheral injection of el-iBoNT can lead to an improvement in central pain-processing processes associated with nerve injury.

## Sensory nerve targets of el-iBoNT in skin sections

Despite existing evidence of analgesic potential of native paralysing BoNT/A (Attal et al, 2016; Zhang et al, 2016; Luvisetto, 2021), there is a paucity of data on targeting of skin sensory neurones by this powerful nerve toxin, which may be due to low, subparalytic doses used. We set out to investigate the possible biological targets of non-paralytic el-iBoNT at 20 ng in the treated area of the rat paw using an antibody targeting botulinum-cleaved SNAP25 (Andreou et al, 2021). Our immunostaining experiments demonstrated botulinum SNAP25-cleaving activity in nerve fibres in paw skin sections after injection of el-iBoNT, and this cleaved SNAP25 staining was largely localised to the dermis (Fig 4, red). Colocalisation experiments demonstrated that cleaved SNAP25 often colocalised with TRPV1; however, this colocalisation was only partial (Figs 4A and S6C and D). In addition, cleaved SNAP25 was found to be present within some NF200-positive fibres in the dermal layer of the skin (Fig 4B). Taken together, these observations suggest that cleaved SNAP25 is present within larger myelinated nociceptors that terminate within the dermis rather than unmyelinated C nociceptors that penetrate the epidermis (Fig 4A and B). In addition, examination of fibres surrounding blood vessels within skin sections revealed

**Figure 2. Functional testing of botulinum molecules.**
**(A)** Immunoblot (left) and graph (right) showing SNAP25-cleaving activity of the native BoNT/A and two novel botulinum molecules. Cleaved SNAP25 (cSNAP25) migrates slightly faster than native SNAP25. Rat DRG neurones were treated with botulinum molecules at indicated concentrations for 68 h before performing immunoblotting using an anti-SNAP25 antibody with an anti-syntaxin antibody serving as a loading control. The proportion of SNAP25 that had been cleaved was determined by band densitometry (n = 3, two-way ANOVA with Tukey's multiple comparisons test). **(B)** Schematic showing detection of botulinum-cleaved SNAP25 by cSNAP25 antibody (left). Immunocytochemical images showing a similar degree of cleavage of SNAP25 (red) by el-iBoNT and native BoNT in rat DRG neuronal cultures. The DRG neurons were treated with either BoNT/A or el-iBoNT at 1 nM concentration, for 24 h, followed by immunostaining for beta-III tubulin (green) and cSNAP25 (red). Scale bars: 100 $\mu$m. **(C)** Graph showing changes in compound muscle action potentials recorded in rat gastrocnemius muscle after subcutaneous injections of two iBoNT molecules (10 ng). The electromyography data obtained 72 h post-injection show that el-iBoNT elicits significantly less motor deficit compared with non-elongated iBoNT ($P < 0.001$, n = 4, two-way ANOVA, Tukey's post hoc test). **(D)** Immunohistochemical examination of cleaved SNAP25 (cSNAP25, red) in rat gastrocnemius muscle after subcutaneous injections of the two iBoNT molecules (10 ng) reveals reduced cleavage of SNAP25 at neuromuscular junctions (NMJs) in the case of elongated iBoNT (scale bar: 50 $\mu$m). Bungarotoxin staining (green) delineates NMJs. The bar chart shows the percentage of NMJs carrying cleaved SNAP25. The percentage of NMJs with cSNAP25 is significantly reduced in animals injected with el-iBoNT compared with non-elongated iBoNT ($P < 0.01$, n = 4, one-way ANOVA, Tukey's post hoc test).

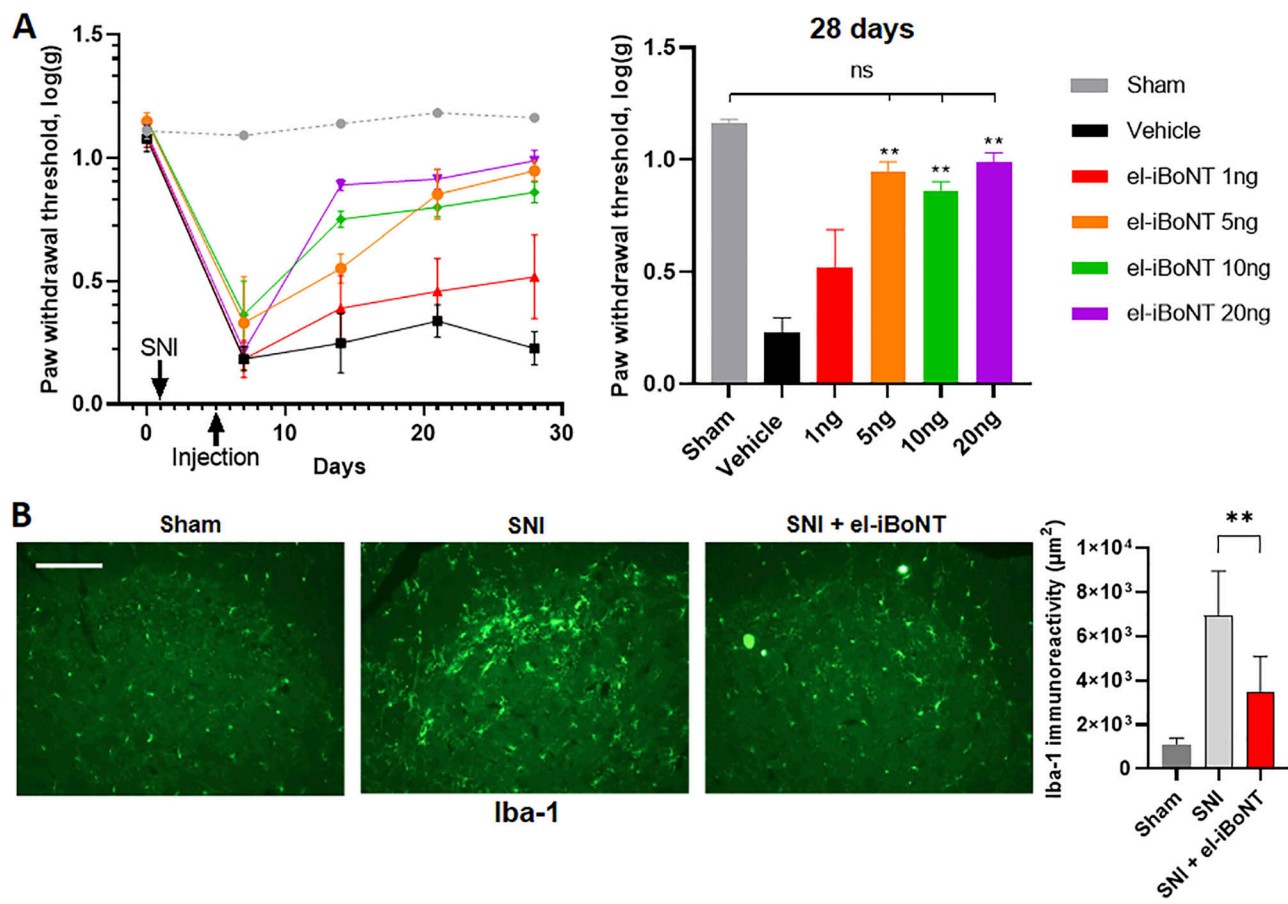

**Figure 3.   Dose-dependent effects of intraplantar injections of el-iBoNT in the rat spared nerve injury (SNI) model.**
Basal paw withdrawal thresholds were measured at day 0, and SNI was performed on day 1. The sciatic nerve was exposed but not manipulated to act as a sham control (grey). On day 5, a single dose of el-iBoNT at indicated amounts or a saline vehicle was administered by intraplantar injection. **(A)** Time course of mechanical hypersensitivity was assessed by recording paw withdrawal thresholds using the von Frey hair stimulation (left). Robust mechanical hypersensitivity was observed in all animals after SNI surgery in comparison with the sham controls. Injections of el-iBoNT reversed mechanical hypersensitivity of the paw because of nerve injury. Data show log of the mean of the 50% threshold ± SEM. The bar chart shows that at day 28, even the 5-ng group exhibited mechanical sensitivity comparable to sham-operated animals, in contrast to the vehicle-only group (two-way ANOVA with Tukey's multiple comparisons test, **$P < 0.01$, $n = 4$) (right). **(B)** Immunostaining for Iba-1 reveals microglial activation in rat spinal cord two weeks after SNI injury (central panel), but not in the sham group (left panel). After a single intraplantar injection of el-iBoNT (20 ng), the number of activated spinal microglia, one week after nerve injury, is significantly reduced (right panel and bar chart, right panel, unpaired $t$ test, *$P = 0.016$) (scale bar: 50 $\mu$m).

colocalisation of cleaved SNAP25 with tyrosine hydroxylase (TH), a marker of a sympathetic noradrenergic subset of neurons (Fig 4C).

## Discussion

Treatment of chronic pain caused by nerve injury remains a major unsolved medical problem. At present, the native unmodified BoNT/A with highly paralysing properties is being evaluated in chronic pain conditions. However, the degree of pain relief achieved with the permissible doses of paralysing native BoNT/A is often smaller than desired. A clinical trial demonstrated a significant but rather modest (20–30%) drop in pain scores in neuropathic pain sufferers after injections of Botox, lasting several months (Attal et al, 2016). It was not possible to inject larger doses to enhance the observed analgesic effects because of the paralytic

properties of BoNT/A. Thus, it would be important to develop BoNTs, which do not paralyse and still have ability to target sensory neurons, so larger doses could bring stronger pain relief. The new results presented here are fourfold: first, we demonstrated here the potential of an isopeptide conjugation system (Zakeri et al, 2012) to make functional botulinum neuronal modulators from two independently produced non-paralytic parts. This technique presents a safe, streamlined approach for manufacturing botulinum molecules for therapeutic use. Second, we show that an el-iBoNT molecule exhibits greatly reduced paralytic ability when directly compared to the non-elongated version, highlighting the importance of structure in the action of BoNT/A-derived molecules. Third, el-iBoNT effectively alleviates nerve injury–induced pain in rats in the low nanogram range, an important improvement over previously described stapled botulinum molecules (Darios et al, 2010; Ferrari et al, 2011; Mangione et al, 2016). Fourth, our immunostaining experiments revealed for the first time the subpopulations of

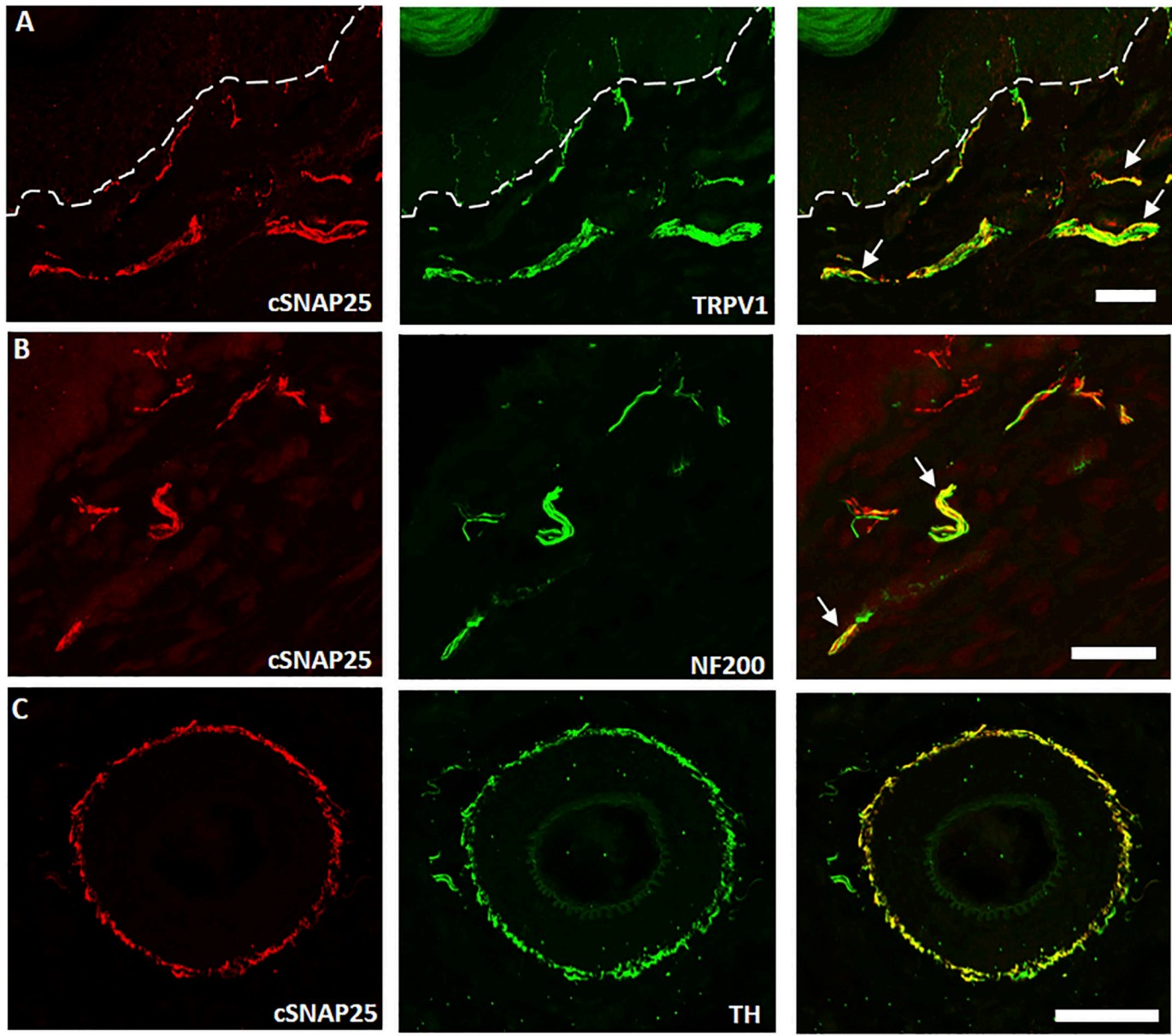

**Figure 4. Localisation of cleaved SNAP25 (cSNAP25) in cutaneous tissue injected with el-iBoNT.**
Colocalisation of cSNAP25 and TRPV1 in dermal fibres of rat cutaneous tissue. **(A)** TRPV1 (green) staining seen to coexist (yellow) with cSNAP25 (red) in dermal but not epidermal tissue. Dermal–epidermal boundary shown with a dotted line. Examples of costaining indicated by arrows. **(B)** NF200, a marker for larger A fibres (green), colocalises with some cSNAP25-positive fibres (red). **(C)** Dermal arteriole costained with tyrosine hydroxylase (TH, green) and cSNAP25 (red) showing a robust colocalisation of the two markers in sympathetic nerve fibres. Scale bars: a, e, f = 50 $\mu$m, and b, c, d = 10 $\mu$m.

afferent nerve fibres targeted by a botulinum type A molecule. Together, our study indicates that a precisely engineered el-iBoNT molecule may provide a new therapeutic option to chronic pain sufferers. This is of great importance considering that there are very limited therapeutic options currently available for chronic pain, and these often lack efficacy, bring about intolerable side effects, and fuel the opioid crisis (King, 2019).

We previously used a protein stapling technique to produce a lesser paralytic BoNT/A (stapled Bitox; Fig S1A) (Darios et al, 2010; Ferrari et al, 2011). This stapling method required production of

three polypeptides, and the stapled molecule alleviated pain only at high nanogram amounts, which may bring about immune response with repeated applications (Mangione et al, 2016). Another disadvantage of the stapling method is that the link between botulinum parts is not covalent, and thus, their dissociation cannot be ruled out, impeding regulatory approval. The possibility of immune complications exists for all non-human proteins, and el-iBoNT was developed to reduce the dosages in comparison with our original stapled Bitox. The low nanogram amounts injected locally would not be too different compared with native BoNT/A injections

(2–10 ng is injected in some cases, e.g., migraine). An additional advantage of botulinum type A injections is that they need to be repeated only after 3–5 mo, and such long intervals reduce the risk of immune responses, especially after local injections of low nanogram amounts. Recently, production of BoNT from two parts using a sortase catalytic reaction has also been demonstrated (Zhang et al, 2017). Similar to stapling, the sortase approach requires an addition of a third element—the sortase—to facilitate formation of BoNT from two parts. However, sortase reactions are reversible in the presence of the enzyme, and therefore, full removal of the enzyme requires additional obligatory steps (Antos et al, 2016). Here, we presented a convenient and streamlined method to generate functional botulinum molecules for therapeutic applications using an isopeptide bonding, which occurs in a spontaneous manner upon mixing of two botulinum parts.

The natural target of native BoNT/A is the NMJ, where it is internalised into the nerve endings via receptor-mediated small vesicle endocytosis and then cleaves SNAP25—an essential part of neurotransmitter release machinery (Pirazzini et al, 2017). The blockade of acetylcholine release results in muscle paralysis, which is a major limitation of using native BoNT/A as a pain therapeutic. However, by structurally elongating the botulinum molecule, we achieved significantly reduced motor deficit while alleviating nerve injury pain. The BoNT/A light-chain metalloprotease and its association with the translocation domain were designed to remain intact in our engineered BoNT, minimising the possibility of interference with the translocation mechanism, which is supported by the picomolar range of EC50s for cleavage of SNAP25 in DRG neuronal cultures (196 versus 584 pM for BoNT/A and el-iBoNT, respectively) (Fig 2A and see the Materials and Methods section). It is possible that the elongated molecule is restricted in its ability to enter the narrow NMJs covered by Schwann cells, or cannot function properly in small synaptic vesicles. Fig S1B demonstrates that el-iBoNT, in contrast to native BoNT/A, may not fit into small synaptic vesicles operating in fast-conducting neurons, which would not apply to larger peptidergic vesicles operating in nociceptive pathways (Lenzi et al, 1999; Merighi, 2018). Interestingly, although the elongation did not significantly affect the SNAP25 proteolytic activity of el-iBoNT in DRG cultures (Fig 2A and B), we observed a strongly reduced SNAP25-cleaving activity in a highly sensitive SiMa neuronal model (Fernández-Salas et al, 2012) (Fig S7), suggesting that the elongation of BoNT has a differential effect on different types of neurons.

It was previously shown that injections of native BoNT/A at subparalytic amounts could not attenuate mechanical hypersensitivity in the rat SNI model (Wang et al, 2017), suggesting that larger doses of botulinum activity are required to achieve analgesia in this robust model of nerve injury pain. In the present study, we show that injections of nanogram amounts of el-iBoNT can reverse mechanical hypersensitivity after a nerve injury in rats without adverse paralytic effects, making it a promising candidate for human clinical research. Our ability to inject larger doses of non-paralytic botulinum molecules without lethal effects allowed us to delineate the target peripheral nerve fibres (Fig 4). One possible mechanism by which BoNT/A may exert its analgesic effect is by preventing peripheral neurotransmitter and pro-inflammatory mediator release from primary afferent nerve endings (Aoki,

2005). Previously, treatment with the stapled Bitox reduced secondary mechanical hyperalgesia (mediated by A-nociceptors), associated with either complete Freund's adjuvant–induced joint inflammation or capsaicin injection, and also the hypersensitivity associated with SNI (Mangione et al, 2016). Yet, C-nociceptive signalling function was not impaired (Mangione et al, 2016). Various studies have previously suggested that there is a functional link between BoNT/A- and TRPV1-containing nerve fibres (Dussor et al, 2014; Ferrandiz-Huertas et al, 2014; Luvisetto et al, 2015; Li & Coffield, 2016; Meng et al, 2016; Zhang et al, 2016; Fan et al, 2017) and the present immunohistochemical data would support this, showing, for the first time, that a BoNT/A-derived molecule cleaves SNAP25 in TRPV1-positive subpopulations of sensory nerve fibres within the skin. Our data also indicate an association of the el-iBoNT with NF200-positive fibres in the dermis and tyrosine hydroxylase–immunopositive sympathetic nerve fibres innervating blood vessels. Future studies shall investigate the levels of neurotransmitters in the skin after el-iBoNT injections and their effect on vasodilation.

The decreased activation of microglia in the ipsilateral dorsal horn was unexpected. Activated microglia have been implicated in the generation of mechanical hypersensitivity after nerve injury, and microglial activation requires up-regulation and release of colony-stimulating factor 1 from the central axons of damaged DRG neurons (Guan et al, 2016) within the dorsal horn. How intact A-fibres treated with el-iBoNT can suppress central microglial activation, presumably caused by damage to DRG neurons, remains unclear, but it may be of importance in explaining the remote effects of BoNT/A on the alleviation of migraine attacks. Indeed, multiple injections of BoNT/A into the scalp, far removed from the meningeal site at which pain is thought to originate, can have a preventative influence on subsequent migraine attacks. The full mechanisms by which current BoNT/A and el-iBoNT provide pain relief are not yet clear, warranting further investigations. It will be also interesting to evaluate the systemic presence of native BoNT/A and el-iBoNT after local administration using ultrasensitive techniques, for example, based on single-molecule arrays, which can detect as little as 200 fg/ml of BoNT/A in blood plasma (Dinh et al, 2017).

The presented approach to produce novel nerve blockers described here will allow researchers to rationally modify non-toxic parts of botulinum toxins, which can subsequently be joined together to form optimised neuronal blockers for therapeutic purposes. Manufacturing of BoNTs is highly restricted, which greatly slows down progress in developing new nerve blockers. Notably, molecular engineering of antibodies, which are of the same size as BoNTs, in the last 30 yr has yielded over 70 new, monoclonal antibodies approved for use in human medicine. In contrast, the highly dangerous nature of BoNTs prevented such pharmaceutical growth and current medicine still relies on the original bacterial molecules. Introduction of a simple and safe way to make a variety of botulinum molecules will aid the botulinum field and pave the way for development of long-sought, safe nerve blockers. Of note, the novel paralysing iBoNT developed here (Figs 1, 2C, and S4A) could provide an interesting alternative to native BoNT and should be explored in various muscle-relaxing applications. The isopeptide bonding approach will enable generation of various combinations of botulinum enzymes and their binding domains to achieve

tailored targeting of neuronal subpopulations not only for pain relief, but also for treatment of other persistent neurological disorders such as Parkinson's disease or epilepsies. The small size of SpyTag also allows generation of neuropeptide-targeted botulinum enzymes when the SpyTag is linked to desired neuropeptides or other targeting moieties. In conclusion, the isopeptide-based bonding of botulinum parts enables production of safe neuronal modulators lacking the adverse effects of muscle paralysis and opens a new avenue for developing treatments for chronic neurological conditions including a major human disorder—neuropathic pain.

# Materials and Methods

## Study design

The purpose of this study was to engineer a non-paralytic nerve blocker for treatment of nerve injury pain. We used the SpyCatcher–SpyTag pair for isopeptide bonding of BoNT parts, which were prepared by expression in *E. coli*. We demonstrated the functionality of engineered botulinum proteins in rat DRG neuron cultures by Western blotting. We determined that an elongated botulinum molecule has reduced paralytic properties as demonstrated in EMG experiments in rats and by observation. We investigated whether this non-paralytic molecule can reduce pain by measuring mechanical hypersensitivity in the SNI model in rats. The localisation and nature of botulinum targets in the rat skin sections were determined by immunocytochemistry. Sample sizes were determined on the basis of previous behavioural, electrophysiological, and molecular data published by our laboratories (Mangione et al, 2016; Andreou et al, 2021). The biological replicate number is noted in all figure legends. Western blotting, EMG, and immunohistochemical imaging data were reproduced in at least four rats.

## Animals

All protocols were performed under the relevant UK Home Office Licence with local ethical approval and in accordance with current UK legislation as defined in the Animals (Scientific Procedures) Act 1986. The guidelines for pain research in animals published by the International Association for the Study of Pain (Zimmermann, 1983) were followed. Animal numbers were based on power calculations using data from previous experiments employing similar protocols. Experiments were carried out in male Sprague Dawley rats aged 6–10 wk. Rats were housed on a 12/12-h light/dark cycle with food and water available ad libitum, and an ambient temperature of 22°C.

## Production of isopeptide-bonded BoNTs

Preparation of isopeptide-bonded BoNT involved recombinant production of two structurally independent units: the receptor-binding domain (HC, BoNT/A 874–1,296, UniProt K4LN57) independently fused to the SpyTag peptide (Zakeri et al, 2012); and

the light-chain/translocation domain (LHN [Darios et al, 2010]) fused to SpyCatcher (Zakeri et al, 2012). The recombinant proteins were expressed in the BL21(DE3) strain of *E. coli* as glutathione S-transferase N-terminal fusions. Proteins fused to glutathione S-transferase were purified on glutathione Sepharose beads (GE Healthcare) and eluted from beads in 20 mM Hepes, pH 7.3, and 100 mM NaCl (buffer A) using thrombin (1 U/liter of culture) because of the presence of LVPRGS sequence, the thrombin-nicking site, after glutathione S-transferase. Note the LHN protein also carries the LVPRGS sequence, which is also efficiently nicked by thrombin to separate the light chain from the translocation domain. Further purification was achieved by gel filtration using a Superdex 200 10/200 GL column (GE Healthcare). The protein yield was ~200 µg per litre of bacterial culture. The isopeptide-bonded BoNT was assembled by mixing the two fusion proteins for 2 h at 20°C, each component at 1 µM concentration, in buffer A. The bonding was evaluated by SDS–PAGE and resulted in a high molecular weight protein (iBoNT), which is the sum of LHN–SpyCatcher and SpyTag–HC. For SDS–PAGE, an SDS sample buffer, lacking any reducing agents, was added with subsequent 5-min boiling. Under such harsh conditions, protein complexes normally disintegrate, but covalently bonded proteins will stay linked. After electrophoresis, the gel was stained with Coomassie stain, revealing the bonding of two botulinum parts. To ensure that thrombin fully cleaves the site between the light-chain and HN domain of the iBoNTs, the samples were boiled for 5 min in an SDS sample buffer containing 2.5% β-mercaptoethanol. Under these conditions, the 50-kD light chain was fully separated from the rest of the iBoNT proteins, confirming that the light and heavy chains are kept together only by the disulphide bond. The assembled iBoNT was separated from unreacted parts by gel filtration in buffer A, when required. Preparation of el-iBoNT was identical, aside from using a variant LHN, which was fused to the syntaxin head domain (syx) (syntaxin 1A, aa 2–160, UniProt Q16623) followed by SpyCatcher at the C-terminal end yielding LHN–syx–SpyCatcher. Purified proteins were aliquoted (20 µl aliquots at 3 µM) and stored at −80°C before further experiments. To test stability, 10 µl aliquots of iBoNTs were thawed and incubated at 4°C for 0, 2, 4, 6, 24, 48, and 72 h. After the incubation, samples were analysed by SDS–PAGE. All samples of iBoNT remained intact for up to 72 h at 4°C (data not shown). Native BoNT/A was from Metabiologics, Lot number A042519-01.

## Immunoanalysis of cleavage of SNAP25 in neurons

Rats were euthanised via cervical dislocation under isoflurane anaesthesia. DRG neurons were isolated from the lumbar spinal cord (L3-6) and digested in an enzyme mix containing 1 mg/ml dispase II (D4693, 0.85 U/mg; Sigma-Aldrich) and 0.6 mg/ml Collagenase XI (C7657,1594 U/mg; Sigma-Aldrich) for 80 min. After digestion, DRG neurons were dissociated by trituration and centrifuged in 15% BSA solution to separate cells from debris. After centrifugation, DRG neurons were seeded onto laminin-coated 96-well plates (L2020; Sigma-Aldrich) in full culture media containing 1% heat-inactivated horse serum (26050; Gibco), 1% penicillin–streptomycin (P0781; Sigma-Aldrich), 20 ng/ml NGFβ (SRP4304; Sigma-Aldrich), 1× B27 (17504-044; Gibco), 1% GlutaMAX (35050-061; Gibco), 20 µM uridine (U3003; Sigma-Aldrich), and 20 µM 5'-fluoro-

2'-deoxyuridine (F0503; Sigma-Aldrich) in Neurobasal-A medium (10888; Gibco). For immunoblotting analysis, dissociated DRG neurons were cultured for 2 d before treatment with 0.1, 1, 10, 100, 1,000, or 10,000 pM BoNT/A, iBoNT, or el-iBoNT diluted in culture media. After 68 h, DRG neurones were lysed and the cleaved SNAP25 was revealed by Western immunoblotting as previously described (Darios et al, 2010). Cleavage of SNAP25 in human SiMa neuroblastoma cells was analysed by Western immunoblotting as previously described (Andreou et al, 2021). The percentage of SNAP25 that had been cleaved by the toxins was determined by band densitometry using the band analysis tool of Bio-Rad Quantity One software, which measured the Trace (average band intensity multiplied by the measured area) of each band. For automatic detection of the bands, the background level for the whole image was set using a rolling disc size of 20 on an empty lane in the middle of the blot, and the sensitivity was set to 2.5. EC50 for cleavage of SNAP25 was calculated by fitting a curve with non-linear regression ([log] agonist − variable slope [four parameters]) in GraphPad Prism, yielding apparent EC50 values 196 pM, 4,677 pM, and 584 pM for BoNT/A, iBoNT, and el-iBoNT, respectively.

For immunocytochemical analysis, dissociated DRG neurons were cultured for 2 d before treatment with 1 nM BoNT/A or el-iBoNT diluted in full culture media. After 24 h, cells were fixed with 4% PFA for 10 min, followed by permeabilisation with 2% Triton X-100 in PBS for 15 min. Cells were incubated in a blocking solution containing 2% fish skin gelatine, 0.1% Tween-20, and 2% bovine serum albumin in PBS for 2 h. This was followed by overnight staining with a primary antibody cocktail containing rabbit polyclonal anti-cleaved SNAP25 antibody (in-house, which recognises the SNAP25 epitope TRIDEANQ that gets exposed upon cleavage by the type A botulinum light chain [Fig 2B, left]) and mouse anti-beta-III tubulin (MAB1195; R&D Systems). The secondary antibodies used were Alexa Fluor 488 goat anti-mouse antibody (A11029; Invitrogen) and Alexa Fluor 594 goat anti-rabbit antibody (A11012; Invitrogen). Stained cells were imaged on a Leica inverted epifluorescent microscope.

## EMG of rat gastrocnemius muscle

Rats were anaesthetised with isoflurane (4% induction, 2% maintenance) so that animals were unconscious but still maintained a withdrawal reflex. All animals used for EMG studies were of the same weight and age to reduce variables such as the amount of adipose tissue. All EMG experiments were undertaken by the same investigator at the same time with the investigator measuring and labelling the point of needle placement. Stimulating electrodes were inserted into the plantar surface of the paw, and current was passed through to elicit a withdrawal. Recording electrodes across the gastrocnemius muscle measured the voltage of the muscle contraction as an EMG signal to ascertain whether any paralysis occurred because of the injection of toxin. Stimulating needle electrodes (ELSTM2; Biopac) were inserted perpendicularly into the muscle ~0.5 cm from the fifth lumbar vertebrae on either side. The anode was placed distally, and the cathode was placed proximally to the recording leg. A ground electrode (EL452; Biopac) was placed in the base of the tail. A reference recording needle electrode (EL450; Biopac) was placed over the tendon of the gastrocnemius

muscle, and a recording electrode was placed in the belly of the medial gastrocnemius muscle. CMAP measurements were performed using a Biopac system. A 0.2-ms pulse stimulation was performed with a voltage stimulator (BSLSTMB; Biopac). Supramaximal stimulation was determined for each recording. The amplitude of the CMAP waveform was then measured. Eight recordings per leg were performed, and the largest three recordings were averaged. Baseline CMAP recordings were determined for the gastrocnemius of each hindlimb. Baseline EMG measurements were recorded on day 0, and whereas the animal remained under anaesthesia, 2 ng/30 µl of iBoNT or el-iBoNT (12.1 and 11.4 fmol, respectively), n = 4 per group, was administered subcutaneously above the gastrocnemius muscle. Animals were observed to full recovery and returned to their home cage. Subsequent EMG measurements were recorded on days 1, 2, 3, and 7, and the fold change in CMAP at each time point was calculated based on the baseline CMAP recording for each rat individually. After EMG, animals were euthanised via cervical dislocation and gastrocnemius muscle was collected for immunohistochemistry.

## Rat model of neuropathic pain

The SNI model was performed in male rats according to the established methodology (Decosterd & Woolf, 2000). Briefly, under isoflurane general anaesthesia (5% induction, 2% maintenance), the skin on the lateral surface of the left hindleg was incised and blunt dissection was used to part the biceps femoris muscle to expose the sciatic nerve and its three terminal branches: the sural, common peroneal, and tibial nerves. The tibial and common peroneal branches were tightly ligated using 5/0 silk and sectioned distal to the ligation leaving the sural nerve intact. Care was taken to avoid any contact or stretching of the spared sural nerve. In the sham control animal group (n = 4), the sciatic nerve was exposed with no manipulation. In all animals, the subcutaneous tissues and skin were sutured with 6/0 Vicryl sutures (Ethicon) and animals were left to recover for a minimum period of 4 d.

To evaluate sensory disturbances, paw withdrawal was analysed using von Frey mechanical sensitivity tests. The same investigator undertook all behavioural analysis and was blind to the treatment group. Baseline measurements were undertaken 4 d (not shown) and 1 d before nerve injury. Animals were placed in a plexiglass chamber on an elevated wire grid. After habituation to the behavioural apparatus for at least 1 h, animals were stimulated using a series of calibrated von Frey filaments via the lateral plantar surface of the paw. The threshold was determined using the up-and-down method (Chaplan et al, 1994) and the 50% paw withdrawal threshold calculated. 4 d after SNI, animals (with the exception of the sham group) underwent subdermal injection of either a vehicle (n = 4) or el-iBoNT (1, 5, 10, or 20 ng/30 µl, or 5.7, 28.6, 57.1, or 114.3 fmol, respectively) (n = 4) into the plantar surface of the ipsilateral paw and paw withdrawals were analysed using von Frey filaments.

## Immunohistochemistry of rat tissues

All tissue segments examined and stained were obtained from sites equidistant from the sites of injection. Needle placement was

observed in the tissue sections and sections obtained from that region. Tissues were serially sectioned and mounted, and corresponding slides were used to ensure tissues examined were from the same site between animals. The gastrocnemius muscles from both the ipsilateral and contralateral sides were collected into a formaldehyde solution made from 4% (wt/vol) paraformaldehyde in PBS and stored at 4°C for 24 h after completion of EMG studies. Muscle tissues were then submerged into 30% sucrose solution in 0.1 M phosphate buffer overnight at 4°C for cryoprotection, frozen in the Tissue-Tek O.C.T. compound (Sakura Finetek), and serially sectioned on a microtome cryostat, at 14 µm onto poly-D-lysine–coated glass slides (Sigma-Aldrich Company Ltd). Tissue sections were blocked in PBS containing 0.2% Triton X-100 (PBST) and 10% Normal Donkey Serum (Jackson ImmunoResearch Laboratories) for 1 h at 20°C followed by overnight incubation with rabbit anti-cleaved SNAP25 (in-house, 1:10,000) at 4°C diluted in PBST and 5% Normal Donkey Serum. Sections were then incubated for 90 min at 20°C with Alexa Fluor 594 donkey anti-rabbit (Jackson ImmunoResearch) and alpha-bungarotoxin Alexa Fluor 488 conjugate (Thermo Fisher Scientific) diluted in PBST and 5% Normal Donkey Serum before being coverslipped using Fluoromount-G mounting medium (Thermo Fisher Scientific). Glabrous skin sections from rats were collected into 2% Zamboni's fixative (0.1 mol/l phosphate buffer, pH 7.4, containing 4% paraformaldehyde and 0.2% picric acid) for 24 h at 4°C. After fixation, specimens were cryoprotected as above and embedded longitudinally in the O.C.T. compound. Serial 14-µm sections were cut and thaw-mounted onto poly-D-lysine–coated glass microscope slides. Sections were collected as 20 sets so that each section was 280 µm from the section adjacent in the same set. Sections were left to air-dry for 1 h at 20°C, before storage at –80°C until required for processing. Skin sections were processed for indirect immunofluorescence as above and dual-labelled with antisera to rabbit anti-cleaved SNAP25 (in-house, 1:10,000) and anti-TRPV1 (1:1,000; Alomone), mouse anti-NF200 (in-house, 1:200), and sheep anti-TH (AB1542, 1:1,000; Millipore). The secondary antibodies used were Alexa Fluor 594 donkey anti-rabbit (Jackson ImmunoResearch), Alexa Fluor 488 donkey anti-mouse (Jackson ImmunoResearch), and Alexa Fluor 488 donkey anti-sheep (Jackson ImmunoResearch). All secondary antibodies were diluted in PBST and 5% Normal Donkey Serum and used at a dilution of 1:100. Spinal cord tissue was removed using hydraulic extrusion. A small groove was made longitudinally along the ventral surface of the spinal cord to allow identification of the left and right sides. The spinal cord was placed in 4% paraformaldehyde in PBS and stored at 4°C for 24 h followed by submersion in sucrose for cryoprotection for a further 24 h. The lumbar segment (L3-5) was dissected and embedded into the O.C.T. compound, and 30-µm frozen sections were collected free-floating into 24-well plates containing PBS. Free-floating spinal cord sections were processed for immunofluorescence to identify microglia using an antibody raised against goat anti-Iba-1 (Abcam) and visualised using secondary Alexa Fluor donkey anti-goat 488. Spinal cord tissues were mounted onto SuperFrost Plus glass slides (J1800AMNT; Thermo Fisher Scientific) coverslipped using Fluoromount-G mounting medium (Thermo Fisher Scientific), and imaged on an Olympus FV1000 confocal laser scanning microscope. The immunohistochemical techniques were repeated and

validated in control experiments in which the primary rabbit anti-cSNAP25 antibody was omitted from the solution resulting in no positive signal, indicating the robust nature of this protocol and the antibody.

## Statistical analysis

Statistical testing was performed using GraphPad Prism, version 8. Values are presented as means ± SE. The tests used are reported in the text and figure legends. In all cases, values of $P < 0.05$ were regarded as significant.

# Supplementary Information

# Acknowledgements

This work was funded by Medical Research Council grant MR/K022539/1 (to SP Hunt and B Davletov) and Medical Research Council Confidence in Concept award MC_PC_16058 (to B Davletov). We thank Anna Andreou for sharing her expertise in electromyography and Anton Maximov for testing iBoNT in neuronal cultures. We also thank Ciara Doran, Anna Andreou, Anton Maximov, and Rashid Giniatullin for their feedback on the article.

## Author Contributions

C Leese: investigation, methodology, and writing—review and editing.
C Christmas: conceptualisation, investigation, visualisation, methodology, and writing—original draft, review, and editing.
J Mészáros: investigation, visualisation, and methodology.
S Ward: investigation.
M Maiaru: investigation and methodology.
SP Hunt: conceptualisation, supervision, funding acquisition, project administration, and writing—review and editing.
B Davletov: conceptualisation, supervision, funding acquisition, project administration, and writing—original draft, review, and editing.

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
