## [Reviewer comments · Life Science Alliance]

New botulinum neurotoxin constructs for treatment of chronic pain

Charlotte Leese, Claire Christmas, Judit Mészáros, Stephanie Ward, Maria Maiaru, Stephen P. Hunt and Bazbek Davletov

DOI: 10.26508/lsa.202201631

Corresponding author(s): Dr. Bazbek Davletov (University of Sheffield)

Review timeline:

Submission Date:	2022-07-26
Editorial Decision:	2022-09-20
Revision Received:	2022-12-15
Editorial Decision:	2023-01-16
Revision Received:	2023-03-14
Editorial Decision:	2023-03-15
Revision Received:	2023-03-17
Accepted:	2023-03-20

Scientific Editor: Novella Guidi

Transaction Report:

No Peer Review Process File is available with this article, as the authors have chosen not to make the review process public in this case.

Re: Life Science Alliance manuscript #LSA-2022-01631-T

Dr. Bazbek Davletov
University of Sheffield
Biomedical Sciences
Western Bank
Sheffield S10 2TN
United Kingdom

Dear Dr. Davletov,

Thank you for submitting your manuscript entitled "New botulinum neurotoxin constructs for treatment of chronic pain" to Life Science Alliance. The manuscript was assessed by expert reviewers, whose comments are appended to this letter. We invite you to submit a revised manuscript addressing the Reviewer comments.

Thank you for this interesting contribution to Life Science Alliance. We are looking forward to receiving your revised manuscript.

Sincerely,

B. MANUSCRIPT ORGANIZATION AND FORMATTING:

Re: Life Science Alliance manuscript #LSA-2022-01631-TR

Dr. Bazbek Davletov
University of Sheffield
Biomedical Sciences
Western Bank
Sheffield S10 2TN
United Kingdom

Dear Dr. Davletov,

Thank you for submitting your revised manuscript entitled "New botulinum neurotoxin constructs for treatment of chronic pain" to Life Science Alliance. The manuscript has been seen by the original reviewers whose comments are appended below. While reviewers 1 and 3 continue to be overall positive about the work in terms of its suitability for Life Science Alliance, some important issues remain. Especially the major concern raised by Reviewer 2 related to the impossibility to evaluate the equipotency of compounds tested given the low resolution image of the Western Blots and the lack of loading controls. All these concerns would need to be addressed in the revised version before resubmission. We, thus, encourage you to submit a revised version of the manuscript back to LSA that responds to all of the reviewers' points.

Please note that I will expect to make a final decision without additional reviewer input upon resubmission.

Please submit the final revision within two month, along with a letter that includes a point by point response to all the remaining reviewer comments.

B. MANUSCRIPT ORGANIZATION AND FORMATTING:

Sincerely,

RE: Life Science Alliance Manuscript #LSA-2022-01631-TRR

Dr. Bazbek Davletov
University of Sheffield
Biomedical Sciences
Western Bank
Sheffield S10 2TN
United Kingdom

Dear Dr. Davletov,

Thank you for submitting your revised manuscript entitled "New botulinum neurotoxin constructs for treatment of chronic pain". We would be happy to publish your paper in Life Science Alliance pending final revisions necessary to meet our formatting guidelines.

- please use the [10 author names, et al.] format in your references (i.e. limit the author names to the first 10)
- please add a figure callout for Figure 2D; Figure S4 D-E and please double-check your callouts for Figure S5-you have callouts for panels that are not in the figure or in the legend

Figure Check:

- Figure S1B: looks like the blots here could be pasted in: please provide source data

A. FINAL FILES:

-- High-resolution figure, supplementary figure and video files uploaded as individual files: See our detailed guidelines for preparing your production-ready images,

<https://www.life-science-alliance.org/authors>

B. MANUSCRIPT ORGANIZATION AND FORMATTING:

Sincerely,

4th Editorial Decision

20 March 2023

RE: Life Science Alliance Manuscript #LSA-2022-01631-TRRR

Dr. Bazbek Davletov

University of Sheffield
Biomedical Sciences
Western Bank
Sheffield S10 2TN
United Kingdom

Dear Dr. Davletov,

Thank you for submitting your Research Article entitled "New botulinum neurotoxin constructs for treatment of chronic pain". It is a pleasure to let you know that your manuscript is now accepted for publication in Life Science Alliance. Congratulations on this interesting work.

DISTRIBUTION OF MATERIALS:

Again, congratulations on a very nice paper. I hope you found the review process to be constructive and are pleased with how the manuscript was handled editorially. We look forward to future exciting submissions from your lab.

Sincerely,
